# Immunological Effects of Epigenetic Modifiers

**DOI:** 10.3390/cancers11121911

**Published:** 2019-12-01

**Authors:** Lucillia Bezu, Alejandra Wu Chuang, Peng Liu, Guido Kroemer, Oliver Kepp

**Affiliations:** 1Service anesthésie-réanimation, Hôpital européen Georges Pompidou, AP-HP, 75015 Paris, France; lucilliabe@gmail.com; 2Faculty of Medicine, University of Paris Sud, 94270 Kremlin-Bicêtre, France; alewch@hotmail.com; 3Equipe labellisée par la Ligue contre le cancer, 75000, Paris, France; KUNZhiling@gmail.com; 4Université de Paris, Sorbonne, INSERM U1138, Centre de Recherche des Cordeliers, 75006 Paris, France; 5Metabolomics and Cell Biology Platforms, Gustave Roussy Cancer Center, 94800 Villejuif, France; 6Pôle de Biologie, Hôpital Européen Georges Pompidou, AP-HP, 75015 Paris, France; 7Suzhou Institute for Systems Medicine, Chinese Academy of Medical Sciences, 215123 Suzhou, China; 8Department of Women’s and Children’s Health, Karolinska Institute, Karolinska University Hospital, 171 76 Stockholm, Sweden

**Keywords:** epigenetic modifiers, immunogenicity, cancer

## Abstract

Epigenetic alterations are associated with major pathologies including cancer. Epigenetic dysregulation, such as aberrant histone acetylation, altered DNA methylation, or modified chromatin organization, contribute to oncogenesis by inactivating tumor suppressor genes and activating oncogenic pathways. Targeting epigenetic cancer hallmarks can be harnessed as an immunotherapeutic strategy, exemplified by the use of pharmacological inhibitors of DNA methyltransferases (DNMT) and histone deacetylases (HDAC) that can result in the release from the tumor of danger-associated molecular patterns (DAMPs) on one hand and can (re-)activate the expression of tumor-associated antigens on the other hand. This finding suggests that epigenetic modifiers and more specifically the DNA methylation status may change the interaction of chromatin with chaperon proteins including HMGB1, thereby contributing to the antitumor immune response. In this review, we detail how epigenetic modifiers can be used for stimulating therapeutically relevant anticancer immunity when used as stand-alone treatments or in combination with established immunotherapies.

## 1. Introduction

Epigenetic modifications involve all molecular mechanisms affecting gene expression in a reversible, transmissible, and adaptive way without altering the DNA sequence. Defined in 1942 by Conrad Hal Waddington, epigenetic modifications were first described in the cellular differentiation process [1]. Although all cells of a multicellular organism possess close-to-identical genomes, gene expression varies according to tissue, allowing functional and phenotypical changes from one cell type to another. This fundamental observation explains how environmental factors govern gene activity and allow the expression of differential phenotypes from the same genetic code. Transgenerational epigenetic inheritance may also explain the loss or the gain of inherited phenotypic characteristics that are passed on to subsequent generations [2]. These phenomena are illustrated by biological and physical differences that develop between cloned animals or twins [3,4].

Epigenetic alterations compose a natural and fundamental regulatory system of the genetic information contained in the DNA sequence of normal cells. This program is essential for terminal differentiation processes as well as for the maintenance of homeostasis. Epigenetic modifications occur physiologically at several levels and play a regulatory role in various cellular functions such as transcription, RNA splicing, and nuclear export as well as translation. Three mechanisms are considered to induce epigenetic changes, namely DNA methylation, histone modification, and non-coding RNA-associated gene silencing. DNA methylation occurs in healthy cells via the addition of methyl groups by DNA methyltransferases (DNMT1, DNMT2, DNMT3) on position 5 of cytosine residues at CpG-rich promoter regions in order to silence specific genes (in the case of parental genomic imprinting, repeated, or deleterious genes) [5,6]. Post-translational modifications of histones including methylation, acetylation, phosphorylation, and ubiquitination also impact on the accessibility of chromatin and on transcription [7,8] (Figure 1).

Epigenetic dysregulation can have severe consequences and may play a key role in the manifestation of complex diseases such as cancer by inactivating tumor suppressor genes and by activating oncogenic pathways. Aberrant DNA methylation is the major epigenetic change leading to oncogenesis and can be subdivided into (i) transition of methylcytosine into thymine after deamination leading to DNA mismatch [9], (ii) inactivation of suppressor genes by methylation of CpG islands [10,11], (iii) changes in expression of genomic imprinting, (iv) genome instability after demethylation of repeated sequences and transposon reactivation [12,13]. In addition, modifications to RNA molecules contribute to cellular transformation, even if such changes occur in non-coding functional RNAs. Indeed, microRNAs and other non-coding RNAs post-transcriptionally regulate mRNA transcripts involved in all major cellular processes. Thus, the repression or overexpression of microRNAs targeting tumor suppressor genes or oncogenes, respectively, promotes tumorigenesis [14].

Epigenetic modifications of histone proteins that play a main role in the process of transcription have major repercussions on DNA replication, the detection and repair of DNA damage, and consequently the susceptibly to malignant transformation. In line with this, the loss of acetylation at Lys16 and loss of trimethylation at Lys20 residues of histone H4 associated with DNA repetitive sequences, or allelic deletion of the H2AX variant, changes the interaction between chaperon proteins and chromatin, thus impacting on genomic integrity and eventually the occurrence of cancer [15,16,17]. In addition, epigenetic dysregulations can reinforce preexisting or induce genetic abnormalities such as a decrease in the abundance of the tumor suppressor p53 that leads to the induction of allelic mutations [10,11]. On the contrary, the transformation of cells with the *K-ras* oncogene can have as a consequence epigenetic adaptations including DNA methylation, chromatin remodeling, and histone modification [13].

Epigenetic regulators are strongly interconnected. Thus, the DNA methyltransferase DNMT1 acts synergistically with DNMT3a and b, with histone methyltransferases SUV39H1 and EHMT2 as well as with the histone deacetylase HDAC2 [18]. Nevertheless, this complex network includes some regulatory checkpoints that can be detected, such as the hypermethylation of certain promoters, and can be directly or indirectly targeted by therapeutic agents, such as the DNMT inhibitor 5-aza-2’-deoxycytidine (decitabine) that is used for the treatment of myelodysplasia and acute myeloid leukemia [19,20]. Moreover, epigenetic variations are less stable than genetic modifications and are theoretically reversible.

Epigenetic modifiers exert various anticancer activities including the induction of apoptosis and the inhibition of angiogenesis. However, several studies showed that epigenetic modifiers have immunomodulatory properties, which impact on both innate and adaptive immune responses. They may affect immune effectors at different levels through the upregulation of MHCI and II expression, the production of cytokines, the elevated transcription of immuno-regulatory genes, and the expression of costimulatory molecules [21,22,23,24]. Finally, some groups demonstrated that HDACi may induce immunogenic cell death characterized by calreticulin exposure, ATP production, and HMGB1 release [25].

Interestingly, pharmacological or genetic DNMT inhibition also results in the translocation of the chromatin-binding protein high mobility group box 1 (HMGB1) from the nucleus to the cytoplasm [26,27]. In the nucleus, HMGB1 serves a key role in chromatin opening and gene transcription; once released (first to the cytoplasm and later to the extracellular milieu) HMGB1 ligates TLR4 on dendritic cells and stimulates the presentation of antigens to T lymphocytes [28]. Epigenetic changes are also implicated in the control of T cells exemplified by the finding that the methylation status of IL-4 and INFɣ genes is associated with the activation of CD4^+^ T cells [29,30]. Similarly, the methylation status of CNS2, an intronic regulatory element, improves Foxp3 stability [31]. Altogether, epigenetic agents acting on DNA methylation may exhibit clinical efficacy not only due to the impact on chromatin remodeling but also via modulating gene expression and thus impinging on the activity of immune effectors. 

Thus, epigenetic therapy offers new medical perspectives to control and eradicate tumor cells in clinical routine. 

In this review, we provide an overview on epigenetic modifiers used as stand-alone agents or in combination with antitumor therapies, focusing on their capacity to induce anticancer immune responses.

## 2. Epigenetic Modifiers Used as Single Therapy

### 2.1. Histone Deacetylase Inhibitors (HDACi)

The histone acetylation status depends on the equilibrium between histone acetyltransferases (HAT) and histone deacetyltranferases (HDAC), which add and remove, respectively, acetyl groups on lysine residues. Acetylated histones increase chromatin accessibility and facilitate the binding of transcription factors to DNA sequences. The imbalance between HAT and HDAC in favor of the latter, which manifests in most types of cancer and is associated with an alteration in gene expression [32,33], spurred the clinical development of HDACi with the aim to re-adjust the HAT/HDAC ratio. HDACi can be grouped into four different chemical families according to their structures: Butyric acid derived (such as valproic acid), hydroxamic acid derived (such as suberoylanilide hydroxamic acid (SAHA)), benzamids (such as entinostat) and cyclic tetrapeptides (such as romidepsin). HDACi have effects on cancer cell proliferation and differentiation, and certain HDACi, including vorinostat, romidepsin, belinostat, and panobinostat, have been approved by regulatory agencies for the treatment of T-cell lymphoma and multiple myeloma [34]. Other HDACi are evaluated in clinical trials for the treatment of hematological and solid malignancies. Besides ongoing advancements, HDACi exhibit immunomodulatory activity by controlling cytokine secretion by tumor cells as well as by impacting on macrophage and dendritic cell functions.

#### 2.1.1. Selective Histone Deacetylase Inhibitors

In different models of solid and hematopoietic tumors, the use of selective HDACi targeting class I HDAC (mocetinostat, entinostat, and romidepsin) as single agent elicited beneficial effects on different antitumor effectors, increasing T lymphocyte infiltration or upregulation of MICA/MICB on the tumor cell surface, thus enhancing natural killer (NK) cell activity through an increase in the ligation of the activating receptor NKG2D, which interacts with MICA and MICB [35,36,37,38] (Table 1). Moreover, several cytokines that bridge innate and adaptive immune responses are upregulated by HDACi altogether contributing to anticancer immunity [35,39,40].

#### 2.1.2. Non-Selective Histone Deacetylase Inhibitors

Most HDACi are not specific and target several classes of HDAC. Thus, the antitumor activity of these non-selective HDACi often cannot be clearly associated with the inhibition of a specific set of HDAC. Trichostatin A (TSA) is a class I and II HDACi known to interfere with cell cycle progression in the G1 and G2-M phases, leading to growth arrest and eventually cell death. TSA was ascribed with antitumor activity due to the induction of apoptosis-related genes. 

Most malignant cells evade the immune system due to a loss or dysfunction of the antigen-presenting machinery. HDACi showed relevant immunomodulatory properties at non-apoptotic doses. Several data suggest that HDACi may impact on the regulation of both innate and adaptive immune responses. Indeed, certain HDACi are known to alter dendritic cells function by decreasing the expression of costimulatory molecules, reducing general cytokine secretion and enhancing indoleamine 2,3-dioxygenase (IDO), an immunomodulatory molecule that is produced by activated antigen-presenting cells (APCs) and is responsible for tryptophan catabolism and thus T-cell activation [42,43]. TSA demonstrated the potential to increase antitumor immune responses both in solid cancers and leukemia in vitro via the expression of components belonging to the antigen processing machinery such as MHC class I and class II, facilitating the activation of cytotoxic T lymphocyte (CTL) [21,22,44,45]. Similarly, in a model of neuroblastoma, retinoic acid has been shown to act as a key modulator of the MHCI presentation process [46]. TSA-treated B16 melanoma cells, which were employed as a therapeutic vaccine in a murine model, induced a specific anticancer immune response that depended on enhanced antigen cross-presentation [21].

In addition, TSA (as well as valproic acid, VPA) had the capacity to promote an innate immune response by enhancing NK cell-mediated cytotoxicity, both in carcinoma and leukemia models via the inhibition of HDAC3, a well-known repressor of NKG2D ligands, which is necessary for the recognition and elimination of cancer cells by NK and CD8^+^ CTL [47]. TSA also increases the acetylation of histone H3 and thus decreases the association between HDAC1 at the promoters of MICA and MICB. As a consequence, TSA upregulates MICA/MICB expression on malignant cells, enhancing their susceptibility to the cytotoxicity of NKG2D-expressing lymphocytes [48,49,50,51] (Table 2).

HDACi can also exhibit effects on the polarization of naïve T cells. In mice, dendritic cells treated by HDACi were unable to induce Th1 responses and downstream immune pathway [52].

In a model of hepatocarcinoma, vorinostat and sodium valproate suppressed miRNAs (such as miR-17, miR-18a, miR-19a, miR-20a, miR-93, miR-106b, and miR-889) that target MICA/B. As a consequence, MICA and MICB are upregulated, which improves tumor cell recognition by innate immune effectors and thus the sensitivity to natural killer cell-mediated killing [53,54]. 

Interestingly, vorinostat induced the exposure of calreticulin (CALR) on the surface of childhood brain tumors [55]. The exposure of CALR is a hallmark of immunogenic cell death and facilitates DC-mediated phagocytosis of (parts of) the dying tumor cells and thus tumor antigen transfer to antigen-presenting cells (APC) [63]. This finding suggests that certain HDACi may actively and sustainably stimulate the immune system by inducing immunogenic cell death, resulting in adaptive immune responses and the establishment of immunological memory (Table 2).

Finally, some HDACi target all HDACs inducing several effects on the immune system. For instance, AR42 and panobinostat provoke an upregulation of MHC class I/II and of MICA/MICB on tumor cells, an increase in tumor infiltration by T cells and a decrease in the number of immunosuppressive MDSC [56,57,58,59,60,62]. These findings were evaluated in a phase II clinical trial enrolling Hodgkin lymphoma patients. In this study, a significant decrease of serum cytokines levels and the suppression of T-cell PD-1 expression after the use of panobinostat was reported, further underlining the immune effects of panobinostat [61] (Table 2, Figure 2).

### 2.2. DNMT Inhibitor

DNMT inhibitors (DNMTi) such as 5-azacytidine (azacytidine) and decitabine are the most frequently used epigenetic modulators employed in clinical routine for the treatment of malignant diseases. Synthetized 40 years ago, these agents show an effective anti-metabolic activity on cancer cells, especially in the setting of acute myeloid leukemia (AML). After administration, DNMTi inhibit DNA methylation and silence regulatory genes critical for diverse metabolic circuitries. DNMTi also induce complex biological effects on the immune system as shown in vitro in different models of solid tumors in which decitabine increased the expression of cancer-testis antigens and MHC class I on cancer cells and enhanced tumor cells lysis by CTL [64,65,66,67,68,69] (Table 3).

Azacytidine also exhibited clinically relevant immunomodulatory effects in myelodysplastic syndrome (MDS). In vitro, azacytidine induced the demethylation of the Foxp3 promoter, which in turn decreased the proliferation and the suppressive function of Treg. More interestingly, the number of Treg observed in the peripheral blood of MDS patients was significantly lower in patients responding to 5-azacytidine treatment as compared to non-responders [76]. In a model of NSCLC that commonly exhibits DNA hypermethylation, the upregulation of PD-L1 transcripts and protein was observed upon treatment with 5-azacytidine [77] and similarly an induction of PD-L1, PD-1, PD-L2, and CTLA-4 expression was noticed in a cohort of leukemia treated with decitabine [73]. This important finding allows us to hypothesize that epigenetic modifiers could be combined with immune checkpoint blockers targeting CTLA-4, PD-1, or PD-L1 to potentiate the immune response against cancer (Table 3).

Interestingly, some groups have demonstrated that DNMTi were able to induce HMGB1 release from the nucleus in osteosarcoma and fibrosarcoma models. HMGB1 release is a hallmark of immunogenic cell death, a specific cell death modality which stimulates adaptive anticancer immune responses [63]. This finding implies that the epigenetic status may not only change the interaction of chromatin with chaperon proteins but contribute to antitumor immune response. DNMTi could be used in combination with agents that induce only partial immunogenic cell death, to compensate a lack of HMGB1 release and boost anticancer immune responses [27] (Table 3, Figure 2).

### 2.3. Combination of DNMT and HDAC Inhibitors

Some studies addressed the possibility of combining several epigenetic agents. The combination of DNMT and HDAC inhibitors allowed the restoration of MHC class I molecule expression, as well as reactivation of other parts of the antigen-presenting machinery in a model of HPV16-associated tumors deficient for MHC class I molecules. This upregulation was efficient enough to induce tumor lysis by CTL after tumor antigen recognition [78]. Another immunoepigenetic approach combining promoter demethylation and histone acetylation by means of decitabine and TSA, respectively, revealed induction of MAGE gene family members [79]. Several alternative combinations of DNMTi and HDACi induced the expression of tumor-associated antigens, promoted lymphocyte infiltration of cancers, and promoted immune response against malignant cells [80]. Moreover, some combinations potentiated the expression of PD-1, PD-L1, PD-L2, and CTLA-4 [73,81]. In summary, these data suggest that immunoepigenetic manipulations might be combined with immunotherapies for optimal therapeutic response (Table 4). 

## 3. Epigenetic Modifiers Combined with Immunotherapies

### 3.1. Histone Deacetylase Inhibitors and Immunotherapies

#### 3.1.1. Selective Histone Deacetylase Inhibitors

Increasing evidence suggests that epigenetic agents boost the immune system by enhancing the expression of tumor antigens and cytokines including chemokines involved in the antitumor response. It is thus tempting to speculate, yet needs to be formally proven, that the combination of epigenetic modifiers with immunotherapies such as checkpoint blockers would achieve optimal immune response against tumor cells.

Class I HDAC inhibitors induce PD-L1 and PD-L2 expression in tumors cells. This durable and stable upregulation is facilitated by the histone acetylation of the PD-L1 and PD-L2 genes and was observed in vitro and in vivo in different solid tumor models [85,86]. Tumor-bearing mice receiving class I HDACi combined with PD-1 blockade exhibited a significant reduction in tumor growth and a better overall survival compared to mice receiving single agents [85,86,87,88] (Table 5). 

Entinostat as stand-alone treatment exhibited efficient responses by suppressing Treg and increasing CD8 T cell infiltration into the tumor microenvironment. Combined with IL-2 or peptide vaccine therapies in a model of kidney cancer or in a castration resistant prostate cancer, this immune effect resulted in tumor growth inhibition and improved overall survival [89,91]. HDACi are known to modulate innate antiviral responses. Surprisingly, coadministration of entinostat and an oncolytic booster vaccine suppressed the primary response against the vaccine vector and extended oncolytic activity. However, it enhanced the secondary response against tumor antigen, reduced the frequency of Treg expressing high level of Foxp3, and improved the outcome [90] (Table 5).

#### 3.1.2. Non-Selective Histone Deacetylase Inhibitors

The family of non-selective HDACi is able to increase the expression of tumor-associated antigens. For instance, the treatment of murine melanoma with non-selective HDACi combined with the adoptive transfer of gp100 melanoma antigen-specific T cells improved the antitumor immune response and reduced tumor growth. The immune response to the combination treatment was characterized by (i) an increase in MHC and tumor-associated antigen expression on the tumor cell surface facilitating tumor cells lysis by CTL, (ii) a decrease in Treg in the tumor microenvironment, and (iii) an improved activity and expansion of adoptively transferred T cells [94,95] (Table 6). 

Combination of non-selective HDACi with immune-activating antibodies such as anti-CD40, anti-CD137, or with immune checkpoint blockers allowed researchers to obtain tumor eradication through similar biological mechanisms such as exacerbated phagocytosis of dead tumor cells, the optimization of antigen presentation by APC, an increase in T cells and a decrease of Tregs in the tumor microenvironment [24,60,81] (Table 6).

In myeloma and lymphoma, HDACi treatment was shown to activate the expression of CD20 and CD38 subsequent to histone hyperacetylation and to increase their abundance on the cell surface. HDACi potentiates the therapeutic effects of rituximab and daratumumab to slow down tumor growth. This strategy may be useful in case of daratumumab or rituximab resistance in myeloma or lymphoma diseases [84,96].

### 3.2. DNMT Inhibitor

Few studies evaluated the impact of immunotherapy effect in synergy with DNMTi. In a murine model for HPV16-associated tumor, 5-azacytidine combined with unmethylated CpG oligodeoxynucleotides or with IL-12-producing cellular vaccine demonstrated additive effects improving CD8-mediated immune responses [97]. Decitabine and anti CTLA-4 potentiated the recruitment of innate and adaptive immune effectors [98]. The co-administrated with photodynamic therapy, which leads to the production of reactive oxygen species, vascular damage, and cell death, DNMTi potentiated antitumor effects by inducing the expression of a silence tumor-associated antigen called P1A [99]. Finally, in a phase II clinical trial, the combination of DNMTi with conventional treatment for multiple myeloma (lenalidomide and autologous stem cell transplantation) followed by autologous lymphocyte infusion induced a high immunogenic cancer testis antigen expression in bone marrow or in CD138 cells allowing specific T lymphocytes response. This finding raised the possibility of triggering a protective immune response decreasing the risk of progression in multiple myeloma patients [100] (Table 7).

### 3.3. Combination of DNMT and HDAC Inhibitors

IFNα potentiates the growth inhibitory activity of azacytidine and romidepsin. In a model of colorectal cancer, cotreatment of cells with inhibitors of both DNMT and HDAC combined with IFN type I triggered several hallmarks of immunogenic cell death (calreticulin exposure and HMGB1 release) enhancing the recruitment into tumor bed of dendritic cells and their maturation into antigen-presenting cells thus facilitating tumor cell lysis by CTL [101]. This novel combination promises a new approach for colorectal cancer (Table 8).

## 4. Conclusions

Epigenetic variations are at the origin of changes in gene expression involved in the manifestation of fatal diseases such as cancer. New findings reveal the evidence that epigenetic regulation may also influence the immune system through several pathways. The renewed interest in epigenetic research focuses on pharmacological interventions that reverse epigenetic cancer hallmarks. Such treatments may involve epigenetic modifiers as single agents or combined with various established immunotherapies, as DNMT and HDAC inhibition can result in the release of DAMP and the expression of tumor-associated antigens, respectively. Thus, combination therapies consisting of immunotherapy in conjunction with epigenetic modifiers that increase both the adjuvanticity and the antigenicity of cancer are promising approaches to reactivate T-cell mediated adaptive immunity against cancer and to reinstate tumor immunosurveillance.

## Figures and Tables

**Figure 1 cancers-11-01911-f001:**
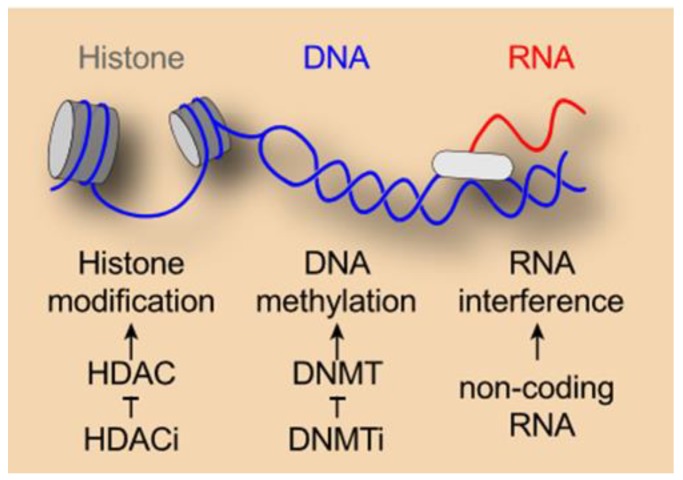
Epigenetic modifications occur at several levels including transcription, translation, splicing, and nuclear RNA release. Several mechanisms are considered to induce epigenetic changes, such as DNA methylation, histone modification, and non-coding RNA-associated gene silencing. Epigenetic modifiers from the group of histone deacetylase inhibitors (HDACi) and DNA methyltransferase inhibitors (DNMTi) can (partially) revert such changes with multiple effects on tumors and the immune system.

**Figure 2 cancers-11-01911-f002:**
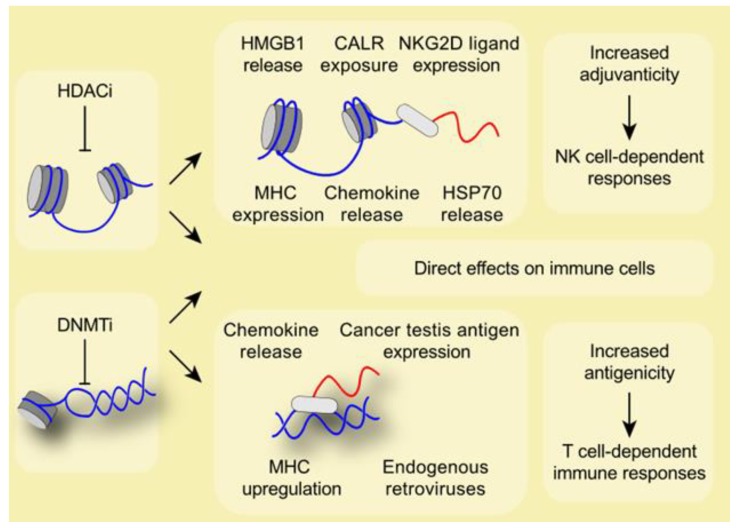
Epigenetic modifiers from the group of histone deacetylase inhibitors (HDACi) affect the accessibility of chromatin and therefore impact on transcription. HDACi induce the release and exposure at the cellular surface of danger-associated molecular patterns (DAMPs) such as high mobility group box 1 (HMGB1), calreticulin (CALR), heat shock protein 70 (HSP70), and a range of chemokines. In addition, the expression of KLRK1 (better known as NKG2D) ligands and major histocompatibility complex (MHC) is increased. DNA methyltransferase inhibitors (DNMTi) affect the antigenicity of tumors by the de novo expression of cancer testis antigens, the availability of MHC molecules, and the concomitant release of chemokines. Independent from tumor specific effects, epigenetic modifiers also exert direct stimulatory effects on immune cells, including macrophages and T cells. Altogether, a combination treatment employing epigenetic modifiers together with immune checkpoint targeting might potentiate the immune response against cancer.

**Table 1 cancers-11-01911-t001:** Selective histone deacetylase inhibitor.

Epigenetic Modifiers	Type of Cancer	Effect on Immune System	Notes	Reference
Mocetinostat, entinostat	Hodgkin lymphoma	Upregulation of CD252 surface expression by inhibition of HDAC11. Repression of the production of IL-10-producing type-1 Treg cells.	In vitro: HL human Hodgkin lymphoma cells.	[39]
Entinostat	Colon neuroblastoma, osteosarcoma, fibrosarcoma	Increased expression of MICA and MICB on tumor cells and NKG2D on primary human NK cells. Enhanced tumor cell lysis.	In vitro: HCT-15 human colon adenocarcinoma cells, COL human neuroblastoma cells, CCH-OS-D human osteosarcoma cells, CCH-OS-T human osteosarcoma cells, HT1080 human fibrosarcoma cells.	[38]
Entinostat	Liver	Enhanced non-specific immune response of exosomes with upregulation of HSP70 and MICB mRNA levels and proteins. Increased NK cell cytotoxicity and peripheral blood mononuclear cell proliferation.	In vitro: Exosomes in HepG2 human hepatoma G2 cells.	[37]
Romidepsin	Lung	Upregulation of multiple T cell chemokines in tumor cells, macrophages and T cells. Upregulation of T cell chemokines, enhanced T cell infiltration and T-cell dependent tumor regression.	In vitro: LKR mouse K-ras mutant lung adenocarcinoma cells. In vivo: LKR cells implanted in 129S4/SvJaeJ mice.	[35]
Romidepsin	Melanoma	Enhanced melanocyte protein Pmel-1 expression in cancer cells promoting tumor specific T-cell-mediated killing of B16/F10 murine melanoma cells. Enhanced CTL-mediated B16/F10 cell killing in vivo.	In vitro: B16/F10 mouse melanoma cells. In vivo: B16/F10 inoculated in C57BL/6 mice.	[36]
Tubastatin A Nexturastat A Mocetinostat	Melanoma	Upregulation of MHC class I expression and melanocyte antigens	In vitro: HEMn-LP, SKMEL21, WM793: Human melanocyte cells, WM164 and WM983A: Two BRAF-mutated melanoma cells In vivo: B16-F10-luc murine melanoma cells injected into C57BL/6 mice.	[41]

CTL, Cytotoxic; T, lymphocyte; HL, Hodgkin lymphoma; IL, interleukine; MICA/B, MHC class I-related chain A/B; NK, natural killer.

**Table 2 cancers-11-01911-t002:** Non-selective histone deacetylase inhibitor.

Epigenetic Modifiers	Type of Cancer	Effect on Immune System	Observations	Reference
TSA	Carcinoma	Increased expression of antigen processing machinery. Enhanced surface expression of MHC I and susceptibility to CTL-mediated killing.	In vitro: TAP-expressing cell line, derived from murine lung cells transformed with HPV16. D11 and A9: TAP-deficient cell line.	[44]
TSA	Melanoma	Enhanced expression of MHC class II, CD40, CD80, and CD86 on B16 melanoma cells. Vaccination with TSA-treated cells induces tumor-specific immunity that involves CD4^+^, CD8^+^ T cells, and NK cells. TSA-treated cells become APCs in vitro and in vivo.	In vitro: B16 mouse melanoma cells. In vivo: Inoculation of mice with TSA-treated B16 cells (vaccination).	[21]
TSA, SAHA, Belinostat	Multiple	Glycogen synthase kinase-3-dependent induction of MHC Class I-related chain A and B on cancer cells, which become targets for NK-cell mediated killing through NKG2D.	In vitro: Jurkat E6-1 human leukaemic T cell lymphoblasts, MCF-7 human breast adenocarcinoma cells, HeLa human cervix carcinoma cells, Daudi human B lymphoblast cells, Aml193 human leukemic cells, Arh77 human plasma cell leukemia cells, DOHH-2 human B cell lymphoma cells, Cem human T cell leukemia cells, Granta human B cell lymphoma cells, U266 human multiple myeloma cells, K562 human chronic myelogenous leukemia cells, HT29 human colon adenocarcinoma cells, DLD-1 human colon adenocarcinoma cells.	[23]
Sodium butyrate, TSA	Neuroblastoma Plasmacytoma Colon	Induction of expression of MHC Class I, II, and CD40 on tumor cells.	In vitro: SK-N-MC human neuroblastoma cells, J558 mouse B myeloma cells, CT26 mouse colon adenocarcinoma.	[22]
TSA	Melanoma	Enhanced expression of genes involved in antigen processing and presentation via the MHC class I pathway. Enhanced cell surface expression of MHC class I, CD40 and CD86 in tumor cells. Increased antigen presentation by tumor cells.	In vitro: B16/F10 mouse melanoma cells.	[45]
TSA	Epithelial tumor	Upregulation of UL16-binding proteins (NKG2D ligands).	In vitro: HeLa, human cervix carcinoma cells HEK 293 human embryonic kidney cells, MCF7, human breast adenocarcinoma, SW480 human colon adenocarcinoma cells, HCT116 human colon carcinoma cells, U937 human histiocytic lymphoma cells, HT29 M6 human colon adenocarcinoma cancer cells.	[47]
TSA	Leukemia	Increased expression of MICA and MICB. Increased tumor cells lysis by NK cell.	In vitro: BALL1 human B cell leukemia cells, Jurkat human lymphoid leukemia cells, K562 human myeloid leukemia cells and patient leukemic cells.	[48]
VPA	Liver	Enhanced expression of MICA and MICB. Increased tumor cells lysis by NK cell.	In vitro: Hep3B human hepatocellular carcinoma cells, HepG2 human hepatocellular cells.	[49]
VPA	Osteosarcoma	Increased MICA and MICB expression. Increased tumor cells lysis by NK cell.	In vitro: Human osteosarcoma cancer cell lines MG-63, HOS, U2OS and SaOS-2.	[50]
VPA	AML	Induction of transcription and expression of NKG2D ligands on tumor cells. Induction of lytic granule exocytosis by autologous CD8^+^ T and NK lymphocytes.	In vivo: In patients with AML.	[51]
Vorinostat	Brain	Induction of CALR exposure in tumor cells.	In vitro: PFSK human neuroectodermal cells and DAOY human medulloblastoma cells.	[55]
Vorinostat	Malignant mesothelioma	Induction of moderate lymphocyte infiltration of tumors. Increased CD8 T cell infiltration.	In vivo: AK7 mouse malignant mesothelioma cells injected in C57BL/6 mice.	[56]
Vorinostat (SAHA)	Breast	Decreased MDSC frequency in the spleen, blood, and tumor bed. Increased proportion of T cells.	In vivo: 4T1 mouse breast cancer cells injected in BALB/c mice.	[57]
TSA, sodium butyrate, VPA	Ovary Cervix	Increased levels of cell surface MICA/MICB in cancer.	In vitro: UCI-101, SKOV-33, Ovcar-3 human ovarian carcinoma cells. HeLa, human cervix carcinoma cells.	[58]
VPA Vorinostat	Prostate	Upregulation of MHC genes.	In vitro: DU145 human prostate cancer cells (an HDACi-sensitive cell line) and PC3 human prostate cancer cells (a relatively HDACi-resistant cell line).	[59]
AR42	Melanoma	Reduction in PD-L1 and PD-L2 expression and ornithine decarboxylase in tumor cells Increased expression of class I MHC. Extracellular release of HMGB1 and HSP70 from tumor cells.	In vitro: TPF-12-293 human melanoma cells.	[60]
Panobinostat	Hodgkin lymphoma	Reduction of serum cytokines levels and suppression of T-cell PD-1 expression.	Phase II clinical trial.	[61]
Panobinostat (LBH589)	Melanoma	Increased expression of MHC class I, MHC class II, and costimulatory molecules CD40, CD80, and CD86.	In vitro: B16 mouse melanoma cells, WM793 and WM983A human melanoma cells. In vivo: B16 cells inoculated in C57BL/6 mice.	[62]

AML, acute myeloid leukemia; APC, antigen-presenting cell; CALR, calreticulin; MHC, major histocompatibility complex; MICA/B, MHC class I-related chain A/B; NK, natural killer; TSA, Trichostatin A; VPA, valproic acid; MDSC, myeloid-derived suppressor cells; TSA, Trichostatin A.

**Table 3 cancers-11-01911-t003:** DNMT inhibitors.

Epigenetic Modifiers	Type of Cancer	Effect on Immune System	Observations	Reference
Decitabine	Ovary	Increased expression of cancer-testis antigens.	In vitro: Human ovarian cancer lines CAOV-3, CAOV-4, COV413, ES-2, OV-90, OVCAR-3, SK-OV-3, SW626, TOV-21G, TOV-112D, and TTB-6, C1R-A2, and C1R-A3.	[64]
Decitabine	Sarcoma	Upregulation of cancer-testis antigens. Enhanced tumor cells lysis by CTL.	In vitro: Rhabdomyosarcoma, osteosarcoma and Ewing’s sarcomas.	[65]
Decitabine	Prostate	Induced expression of a prostate cancer-testis antigen SSX2.	In vitro: LAPC4, MDA-PCa-2b human prostate cancer cells.	[70]
Decitabine	Melanoma	Induced MAGEA1 expression and tumor cell lysis by MAGEA1 specific major histocompatibility complex restricted CTL.	In vitro: 888-mel human melanoma cells.	[66]
Decitabine	Melanoma	Upregulation of MHC class I antigens and of ICAM-1, increased lysis of tumor cells by melanocyte protein Pmel-1 specific CTL with enhanced IFNγ release.	In vitro: Mel 275 human melanoma cells.	[71]
Decitabine	Melanoma	Induction of cancer-testis antigens.	In vivo: Melanoma cells grafted into BALB/c and nu/nu mice.	[67]
Decitabine	Melanoma	Upregulation of HLA-A and -B transcription, cell surface expression of MHC class I antigens, and enhanced tumor cell recognition by MAGE-specific CTL.	In vitro: MSR3-mel human melanoma cells.	[72]
Decitabine	Neuroblastoma	Upregulation of MAGEA1, MAGEA3, and CTAG1B and CTL-mediated tumor cell killing.	In vitro: BE2C, NBL-S, Kelly, NGP, SHSY5Y, EB2M17, IMR32SKN-AS, SKN-SH, SKNMC, CHP134 neuroblastoma cells.	[68]
Decitabine	Leukemia	Upregulation of MAGEA1, MAGEA3, MAGEB2, and CTAG1B. Increased susceptibility of tumor cells to antigen-specific recognition by CTL.	In vitro: U937, human myeloid leukemia, HL60 human promyelocytic leukemia cells, THP-1 human monocytic leukemia cells, and Kasumi-1 human leukemia cells.	[69]
Decitabine	MDS, CMML, AML	Enhanced PD-L1, PD-L2, PD-1, CTLA4 expression in tumor cells.	In vitro: KG-1, HL-60, NB4, THP1, U937, ML1, OCI-AML3, and HEL human acute myeloid leukemia cells and cells from MDS, CMML, and AML patients.	[73]
Decitabine	Lymphoma	Induced CD80 expression in cancer cells that stimulates specific T lymphocyte responses. Infiltration of IFN-γ producing T lymphocytes into tumors.	In vitro: EL4 mouse lymphoma cells. In vivo: EL4 cells injected into mice C57Bl/6.	[74]
Azacitidine	Breast colorectal ovary	Upregulation of IFN signaling, antigen processing and presentation, cytokines/chemokines, and cancer testis antigens.	In vitro: Breast, colorectal and ovarian cancer cells.	[75]
Azacitidine	MDS	Decreased number of Treg and T-helpers in vitro and in patients. Reduced suppressive function of Treg Increased production of IL-17.	In vitro: Treg and T-helpers isolated from MDS patients. Peripheral blood T cells from patients.	[76]
Azacitidine	NSCLC	Upregulation of genes involved in innate and adaptive immunity and PD-L1.	In vitro: NSCLC human non-small cell lung carcinoma cells.	[77]
Azacitidine, Decitabine	Osteosarcoma, fibrosarcoma	Increased plasma HMGB1 levels.	In vitro: U2OS human osteosarcoma cells and MCA205 mouse fibrosarcoma cells. In vivo: C57BL/6 mice.	[27]

CTL, cytotoxic T lymphocyte; ICAM, intracellular cell adhesion molecule-1; IFN, interferon. AML, acute myeloid leukemia; CMML, chronic myelomonocytic leukemia; MDS, myelodysplastic syndrome; NSCLC, non-small cell lung cancer.

**Table 4 cancers-11-01911-t004:** Combination of DNMT and HDACi.

Epigenetic Modifiers	Type of Cancer	Effect on Immune System	Observations	Reference
TSA + Azacytidine	HPV16-associated tumor	Induction of surface re-expression of MHC class I molecules leading to lysis by CTL. Upregulation of antigen-presenting machinery.	In vitro: TC-1/A9, murine tumor cell line expressing the oncogenes E6 and E7 from human papilloma virus 16 and deficient in MHC class I expression.	[78]
Decitabine + TSA	Breast, colorectal	Upregulation of MAGE gene expression.	In vitro: WiDr human colorectal adenocarcinoma cells, MCF-7 human breast adenocarcinoma cells, MDA-MB-231 triple-negative breast cancer cells.	[79]
Decitabine + depsipeptide	Esophagus, pancreas, ovary, mesothelioma, osteosarcoma, lung	Increased expression of tumor antigen CTAG1B on tumor cells, resulting in IFNγ responses by antigen specific T cells.	In vitro: BE-3 human esophageal carcinoma cells, H2373 human pleural mesothelioma cells, Panc-1 human pancreatic cancer cells, OVCAR-3 human ovarian cancer cells, LNZAT3WT4 human osteosarcoma cells, H1299 non-small cell lung carcinoma cells.	[82]
VPA, SAHA, decitabine	MPM	Tumor antigen expression and tumor cell killing by CTL; decitabine + VPA inhibit promote lymphocyte infiltration and enhance T-cell antitumor response in vivo.	In vitro: Human epithelioid mesothelioma cells (established from pleural effusion). In vivo: AK7 murine mesothelioma cells injected into C57BL/6 mice.	[83]
Vorinostat, VPA, panobinostat + entinostat	TNBC	Upregulation of PD-L1 mRNA and protein expression in tumor cells.	In vitro: TNBC triple-negative breast cancer cells.	[81]
VPA + Romidepsin	Lymphoma	Increased CD20 expression.	In vitro: HBL-2 human mantel cell lymphoma cells, TK and B104 human diffuse large B-cell lymphoma cells, Daudi, BJA-B, Namalwa, Raji and Ramos, five human Burkitt lymphoma-derived cells.	[84]
VPA + Hydralazine	Osteosarcoma	Increased expression of cell surface CD95, cell surface MICA, and MICB. Enhanced susceptibility of tumor cells to CD95 and NK cell-mediated cell death.	In vitro: Human osteosarcoma cell lines HOS, U2OS and SaOS-2.	[80]
Vorinostat + Azacitidine	MDS, AML, CMML	Upregulation of PD-L1, PD-L2, PD-1, and CTLA-4 expression.	Phase II clinical trial: CD34+ cells from MDS, CMML, and AML patients.	[73]

AML, acute myeloid leukemia; CLC, chronic myelomonocytic leukemia; CTL, cytotoxic T lymphocyte; HPV, human papilloma virus; IFN, interferon; MDS, myelodysplastic syndrome; MPM, malignant pleural mesothelioma; NK, natural killer; SAHA, suberoylanilide hydroxamic acid; TNBC, triple-negative breast cancer. CMML, chronic myelomonocytic leukemia.

**Table 5 cancers-11-01911-t005:** Selective histone deacetylase inhibitors combined with immunotherapies.

Epigenetic Modifiers + Another Drug	Type of Cancer	Effect on Immune System	Observations	Reference
Entinostat + IL-2 or entinostat + survivin-based vaccine therapy	Kidney Prostate	Reduction of Foxp3 levels in Treg.	In vivo: Murine renal cell carcinoma (RENCA) model or a survivin-based vaccine therapy in castration-resistant prostate cancer (CR Myc-CaP, mouse prostate cancer cells).	[89]
Entinostat + oncolytic virus therapy	Melanoma	Enhanced oncolytic activity of vesicular stomatitis virus, preserved secondary tumor-specific CTL and antibody responses, enhanced viral vector-induced lymphopenia, and reduce Treg.	In vivo: Mice bearing 5-day-old intracranial B16/F10 melanoma.	[90]
Entinostat + IL-2	Kidney	Increased number of CD4^+^ CD25^+^ T cells Decreased number of Treg.	In vivo: Murine renal cell carcinoma (RENCA) luciferase-expressing cells implanted in BALB/c mice.	[91]
Romidepsin + anti PD-1	Lung	Enhanced response to PD-1 blockade and IFNγ-dependent tumor rejection; enhanced activation of tumor-infiltrating T cells.	In vivo: LKRm 13 lung cancer cells injected in 129S4/SvJaeJ mice.	[35]
Depsipeptide + immune cell adoptive transfer therapy	Melanoma	Enhanced CTL-mediated tumor cell lysis; decreased metastatic tumor growth.	In vivo: B16/F10 injected in C57BL/6 mice.	[36]
Mocetinostat + atezolizumab	TNBC	Increased PD-L1 expression.	In vitro: MDA-MB-231, BT-20, MDA-MB-468, BT-549, HS-578T human breast cancer cells.	[86]
Mocetinostat + anti PD-L1	Colon	Increased CD8 cells and decreased Treg cells. Increased anti-tumor activity and clonality of the T-cell repertoire.	In vivo: CT26 mouse colon carcinoma cells injected in BALB/c mice.	[87]
Entinostat + anti PD-1	Lung Kidney	Inhibition of immunosuppressive function of polymorphonuclear- and monocytic-myeloid derived suppressor cells, down-regulation of Treg Increased infiltration of CD8.	In vivo: Renal carcinoma mouse model (RENCA), LLC: Murine Lewis lung carcinoma.	[92]
Nexturastat A + anti PD-1	Melanoma	Enhanced infiltration of immune cells. Increased T cell memory. Reduced pro-tumorigenic M2 macrophages. Neutralized the upregulation of PD-L1 Increased expression of MHC class II.	SM1 mouse melanoma cells.	[88]
Ricolinostat + bromodomain inhibitor JQ1	NSCLC	Enhanced activation of tumor infiltrating CD8 T cells and secretion of effector cytokine IFNγ, Increased CD8/Treg ratio.	In vivo model: Lung tumor induction in mice with Cre-encoding adenovirus intranasally injection.	[93]
Panobinostat + anti PD-1	Melanoma	Upregulation of PD-L1 and PD-L2 expression in melanoma cells.	In vivo: C57BL/6 mice inoculated with B16/F10 melanoma cells.	[85]

CTL, cytotoxic T lymphocyte; IFN, interferon; NSCLC, non–small cell lung cancer; TNBC, triple-negative breast cancer; LLC, Lewis lung carcinoma.

**Table 6 cancers-11-01911-t006:** Non-selective histone deacetylase inhibitors combined with immunotherapies.

Epigenetic Modifiers + Another Drug	Type of Cancer	Effect on Immune System	Observations	Reference
Vorinostat and panobinostat + anti CD40 and anti CD137	Solid tumors	Stimulation of uptake of dead tumor cells by APCs.	In vivo: Mice with established tumors: 4T1.2 (breast), MC38 (colon), or renal (RENCA) murine carcinoma.	[24]
Dacinostat + Pmel-1 immunotherapy	Melanoma	Increased expression of MHC and tumor-associated antigen on tumor cells.	In vivo: B16/F10 mouse melanoma (C57BL/6 mice).	[94]
Panobinostat + Pmel-1 immunotherapy	Melanoma	Reduced Treg. Induction of the expression of IL-2 receptor and the co-stimulatory molecule CD134 in T cells.	In vivo: B16/F10 mouse melanoma (C57BL/6 mice).	[95]
TSA + cytokine induced killer cells	Ovary	Increased expression of MICA and MICB in tumor. Increased antitumor activity of cytokine induced killer cells.	In vivo: UCI-101 implanted subcutaneously into nu/nu mice.	[58]
Vorinostat + anti PD-1	TNBC	Increased T cell and decreased Treg tumor infiltration.	In vivo: Triple-negative 4T1 breast cancer mouse model.	[81]
AR42 + pazopanib	Melanoma	Increased expression of class I MHC molecule and enhanced HMGB1 and HSP70 release.	In vivo: MEL28-R human melanoma tumors isolated from mice.	[60]
AR42 or VPA + anti PD-1	Melanoma	Increased levels of CCL2, CCL5, CXCL9, and CXCL2. Improved activated T cell, M1 macrophages, neutrophils, and NK cell infiltration.	In vivo: B16 mouse melanoma model.	[60]
Panobinostat + daratumumab	Myeloma	Increased CD38 expression and antibody-dependent cellular cytotoxicity.	In vitro: MM1.S human myeloma cells.	[96]
VPA + Rituximab	Lymphoma	Increased cytotoxicity activity of rituximab through upregulated expression of CD20 by VPA.	In vivo: BJA-B cells injected in non-obese diabetic immunodeficiency (NOD/SCID) mice.	[84]

APC, antigen-presenting cell; CTL, cytotoxic T lymphocyte; HMGB1, high mobility group box 1; HSP, heat shock protein; MHC, major histocompatibility complex; NK, natural killer; TNBC, triple-negative breast cancer; VPA, valproic acid.

**Table 7 cancers-11-01911-t007:** DNMT inhibitors combined with immunotherapies.

Epigenetic Modifiers + Another Drug	Type of Cancer	Effect on Immune System	Observations	Reference
5-azacytidine + non-specific immunotherapy CpG oligodeoxynucleotides or IL-12-producing cellular vaccine	HPV16 associated tumors	Induction of CD8 cell-dependent mechanisms. Increased cell surface expression of MHC I, antigen-presenting machinery and IFNγ-signaling pathway.	In vivo: TC-1/A9 tumors cells (HPV16-associated tumors) transplanted into C57BL/6 mice.	[97]
Decitabine + anti CTLA-4	Ovary	Upregulation of chemokines recruiting NK and CD8 T cells, enhancing production of IFNγ and TNFα.	In vivo: BR5FVB1-Akt mouse epithelial ovarian cancer cells inoculated into FVB mice.	[98]
Decitabine + PDT	Lung, Breast, Colon	Induced expression of a silenced tumor-associated antigen P1A.	In vivo: Lewis lung carcinoma, 4T1 and EMT6 mouse mammary carcinoma, CT26 mouse colon carcinoma.	[99]
Azacitidine + Lenalidomide + autologous stem cell transplantation	Multiple myeloma	Upregulation of cancer testis antigens inducing specific T cell response.	Phase II clinical trial.	[100]

IFN, interferon; MHC, major histocompatibility complex; NK, natural killer; PDT, photodynamic therapy; TNF, tumor necrosis factor.

**Table 8 cancers-11-01911-t008:** DNMT and HDAC inhibitors combined with immunotherapies.

Epigenetic Modifiers + Another Drug	Type of Cancer	Effect on Immune System	Observations	Reference
5-azacytidine + entinostat + anti PD-1/anti CTLA-4	Breast	Circulating MDSC decrease.	In vivo: 4T1 tumor-bearing mice.	[102]
Azacitidine + romidepsin + IFNα	Colorectal	CALR translocation, HMGB1 release, DC-mediated phagocytosis of drug-treated cancer cells.	In vitro: SW620, CTSC#18 colorectal cancer cells. In vivo: SW620, CTSC#18 developing in NOD-SCID mice.	[101]

CALR, calreticulin; DC, dendritic cell; HMGB1, high mobility group box 1; IFN, interferon; MDSC, myeloid-derived suppressor cell.

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
