# Peer review of "Immunological Effects of Epigenetic Modifiers"

_cancers, 2019, doi:10.3390/cancers11121911_

Round 1
Reviewer 1 Report
The authors discussed mostly the published immunological effects of single use of epigenetic drugs and in combination with the other epigenetic-targeting drugs or immunotherapies in the treatments of various cancer types. This review article may provide readers an overview of potential molecular mechanisms involved in regulation of immune responses in vitro and in vivo. However, there are several items will need to be improved and corrected before further consideration.
The abstract needs to be reworked. The current version focused on immunity changes with alterations of DNA methylation. Since the authors mentioned different kinds of epigenetic modifiers, a wider implication for abstract should be considered. Some citations were missing: page3, line 104. The authors should put a reference citation at the end of Foxp3 expression. Page3, line 124. Should be a reference at the end of T-cell lymphoma and multiple myeloma. Tables needs to be re-edited. Especially the layout of “effect on immune system”. The authors should simplify it.
Author Response
The authors discussed mostly the published immunological effects of single use of epigenetic drugs and in combination with the other epigenetic-targeting drugs or immunotherapies in the treatments of various cancer types. This review article may provide readers an overview of potential molecular mechanisms involved in regulation of immune responses in vitro and in vivo. However, there are several items will need to be improved and corrected before further consideration.
The abstract needs to be reworked. The current version focused on immunity changes with alterations of DNA methylation. Since the authors mentioned different kinds of epigenetic modifiers, a wider implication for abstract should be considered.
We thank the reviewer for this pertinent remark. Following her/his advice we have improved the abstract of the revised manuscript.
Some citations were missing: page3, line 104. The authors should put a reference citation at the end of Foxp3 expression. Page3, line 124. Should be a reference at the end of T-cell lymphoma and multiple myeloma.
We have added the requested references.
Tables needs to be re-edited. Especially the layout of “effect on immune system”. The authors should simplify it.
The referee is right. We have re-edited and simplified the tables with a special focus on “ effect on immune system”
Reviewer 2 Report
This review focuses on describing how epigenetic effectors can be used to control immune responses in oncogenesis. The authors describe the effects of HDAC and DNMT inhibitors on the immune system when used as stand-alone treatments or in combination with established immunotherapeutic drugs. The authors present a plethora of information about how HDACi and DNMTi affect the immune response in multiple cancer cell lines and clearly depict the systems used to decipher all these pieces of information. The review could be valuable to scientists needing this kind of information. However, it feels more like cataloging of drugs and cell lines, rather than critically revisiting the molecular basis of epigenetic modifiers and immunotherapeutic approaches. The review is worth publishing, with the incorporation of highlighted examples that more precisely describe the immunological effects of the epigenetic modifiers. Without these examples, the authors fail to capture the complex and intriguing dynamic interrelations of immunoepigenetics and immunotherapeutics, greatly reducing the enthusiasm for this review. Finally, a similar review (https://doi.org/10.3390/ijms20092241) has been published recently making the information presented in the current review on HDACi inhibitors rather redundant. The abstract highlights the example of HMGB1 dysregulation by epigenetic alterations but substantial information about the role of HMBG1 is missing in the review. The introduction defines epigenetic events and presents main epigenetics regulatory effects, but fails to connect it to immune responses. For someone that is not an expert on cancer immunotherapy, there is a major gap in the immunological effects that the review is about to focus on. The concluding remarks regurgitate the main points of the review without offering any insight on the importance of an epigenetic approach in cancer immunotherapy. Minor points: line 48: nuclear RNA release is more commonly called RNA export line 49: (epigenetic modifications) they seem to depend mostly on transcriptional factors and chromatin organization - this statement is probably historically true as most epigenetic studies focus on the intertwining of TFs and chromatin organization, however the dynamic interaction of all the levels mentioned above including epitranscriptome, splicing, export, and non-coding RNAs probably play equally important roles in regulating epigenetic events. line 52: translocation of methyl groups is not commonly used, addition of methyl groups is better. line 55: chemical post-translational arrangements also sound uncommon, post-translational modifications of histones is the common term in epigenetics. line 96: Apart from... to the cytoplasm - this sentence does not relay a coherent message. What do you mean epigenetic modifiers confer secondary consequences? what epigenetic modifiers provoke the translocation of HMGB1 from the nucleus to the cytoplasm? line 104: what does "conditions the expression of Foxp3 expression" mean? 118: clincial is mistyped. 122: inlcuding is mistyped. 130: elicted is mistyped. 164: vorinostat demonstrated its capacity ... needs to be rephrased. 171: Such HDACi such as AR42 and panobinostat have several effects on the immune system such as an upregulation - needs to be rephrased Figure 2: I am guessing the green dots represent acetyl groups. It would be useful to explain this. I am not sure I understand what the floating green dots represent. DNA methylation is commonly presented in figures with a line and a circle, the circle is black for the presence of a methyl group and the circle is not filled for the absence of methyl group. This may also be useful to be added in Fig.2. 180: the line is bold erroneously. 194: (AML-. The parenthesis is not closed 194: What do you mean after incorporation into DNA, DNMTi inhibit DNA methylation? 281: Table 6 is bold. 300: there are 2 periods.Author Response
This review focuses on describing how epigenetic effectors can be used to control immune responses in oncogenesis. The authors describe the effects of HDAC and DNMT inhibitors on the immune system when used as stand-alone treatments or in combination with established immunotherapeutic drugs. The authors present a plethora of information about how HDACi and DNMTi affect the immune response in multiple cancer cell lines and clearly depict the systems used to decipher all these pieces of information. The review could be valuable to scientists needing this kind of information. However, it feels more like cataloging of drugs and cell lines, rather than critically revisiting the molecular basis of epigenetic modifiers and immunotherapeutic approaches. The review is worth publishing, with the incorporation of highlighted examples that more precisely describe the immunological effects of the epigenetic modifiers. Without these examples, the authors fail to capture the complex and intriguing dynamic interrelations of immunoepigenetics and immunotherapeutics, greatly reducing the enthusiasm for this review. Finally, a similar review (https://doi.org/10.3390/ijms20092241) has been published recently making the information presented in the current review on HDACi inhibitors rather redundant.
We thank the reviewer for this important suggestion. We have incorporated more examples which describe immunological effects of epigenetic modifiers specifically for HDACi.
The abstract highlights the example of HMGB1 dysregulation by epigenetic alterations but substantial information about the role of HMBG1 is missing in the review.
In response to the reviewers request we have added a paragraph on HMGB1 dysregulation by DNMTi to the revised version of the manuscript and discussed the implications of this effect.
The introduction defines epigenetic events and presents main epigenetics regulatory effects, but fails to connect it to immune responses. For someone that is not an expert on cancer immunotherapy, there is a major gap in the immunological effects that the review is about to focus on.
We are grateful to the reviewers constructive critique and have added in the introduction a specific part connecting epigenetic modification with anticancer immune responses.
The concluding remarks regurgitate the main points of the review without offering any insight on the importance of an epigenetic approach in cancer immunotherapy.
In order to address the referee’s concern we have re-edited the concluding remarks section. In the revised version of the manuscript we now point underline the main approach of epigenetic modification in cancer immunotherapy
Minor points: line 48: nuclear RNA release is more commonly called RNA export line 49: (epigenetic modifications) they seem to depend mostly on transcriptional factors and chromatin organization - this statement is probably historically true as most epigenetic studies focus on the intertwining of TFs and chromatin organization, however the dynamic interaction of all the levels mentioned above including epitranscriptome, splicing, export, and non-coding RNAs probably play equally important roles in regulating epigenetic events. line 52: translocation of methyl groups is not commonly used, addition of methyl groups is better. line 55: chemical post-translational arrangements also sound uncommon, post-translational modifications of histones is the common term in epigenetics. line 96: Apart from... to the cytoplasm - this sentence does not relay a coherent message. What do you mean epigenetic modifiers confer secondary consequences? what epigenetic modifiers provoke the translocation of HMGB1 from the nucleus to the cytoplasm? line 104: what does "conditions the expression of Foxp3 expression" mean? 118: clincial is mistyped. 122: inlcuding is mistyped. 130: elicted is mistyped. 164: vorinostat demonstrated its capacity ... needs to be rephrased. 171: Such HDACi such as AR42 and panobinostat have several effects on the immune system such as an upregulation - needs to be rephrased
We thank the reviewer for all these points and we have rephrased and corrected all mistyping
Figure 2: I am guessing the green dots represent acetyl groups. It would be useful to explain this. I am not sure I understand what the floating green dots represent. DNA methylation is commonly presented in figures with a line and a circle, the circle is black for the presence of a methyl group and the circle is not filled for the absence of methyl group. This may also be useful to be added in Fig.2.
We thank the reviewer for pointing this out. We have removed the green dots to ease the understanding of the figure.
180: the line is bold erroneously. 194: (AML-. The parenthesis is not closed.
We have reformatted and edited accordingly.
194: What do you mean after incorporation into DNA, DNMTi inhibit DNA methylation?
We have changed “after incorporation into DNA” to “after administration” in order to be more understandable
281: Table 6 is bold. 300: there are 2 periods.
We have corrected the formatting.
Reviewer 3 Report
The review article gives a good insight into the world of epigenetic drugs and their role in cancer and effects on immune system and related gene expression. The review would become much more attractive if the authors would extend the initial paragraph to include more details on epigenetic regulation of immune system. Few minor details that should be corrected.
a number of recent publications has shown that non-coding RNAs can mediate gene silencing as well as gene activation. This issue should be taken into consideration in the review article and appropriately needs to be discussed. Nuclear RNAi is not well defined in mammalian cells and not very effective as a tool for silencing nuclear non-coding RNAs. This fact needs to be considered in Figure-1. Figure-1, HDACi blocks HDACs not histone modifications, needs change. Line 52, “translocation of methyl groups by DNA methyltransferases (DNMT1, DNMT2, DNMT3)” role of DNMT2 in mammalian cells in DNA CpG methylation is not supported by data. Lines, 70,71 “cell transformation after gene expression disturbance by parental genomic imprinting”- this needs change. ‘Loss’ of genomic imprinting in some instances leads to cell transformation. Imprinting per se does not cuase gene expression disturbance.It is not clear what authors mean by the following and need changes
Lines, 71, 72 “improvement of transposon reactivation” Line 72 “epigenetic alterations of RNA sequences”Author Response
The review article gives a good insight into the world of epigenetic drugs and their role in cancer and effects on immune system and related gene expression. The review would become much more attractive if the authors would extend the initial paragraph to include more details on epigenetic regulation of immune system.
We thank the reviewer for this remark and we have added in the introduction a specific part connecting epigenetic modifier and immune responses.
Few minor details that should be corrected. A number of recent publications has shown that non-coding RNAs can mediate gene silencing as well as gene activation. This issue should be taken into consideration in the review article and appropriately needs to be discussed.
We have completed our review with a paragraph dealing with epigenetic modifiers effects on non-coding RNAs and their impact on anticancer immune responses. We refrained from discussing non-immunological effects of epigenetic modifiers on non-coding RNA because this would have been largely out of scope of the present review.
Nuclear RNAi is not well defined in mammalian cells and not very effective as a tool for silencing nuclear non-coding RNAs. This fact needs to be considered in Figure-1. Figure-1, HDACi blocks HDACs not histone modifications, needs change.
In response to this point, we have modified the figure 1 accordingly.
Line 52, “translocation of methyl groups by DNA methyltransferases (DNMT1, DNMT2, DNMT3)” role of DNMT2 in mammalian cells in DNA CpG methylation is not supported by data.
Here we have added a reference that supports this point.
Lines, 70,71 “cell transformation after gene expression disturbance by parental genomic imprinting”- this needs change.
‘Loss’ of genomic imprinting in some instances leads to cell transformation. Imprinting per se does not cause gene expression disturbance.
In response to the reviewers concern, we have rephrased the sentence
It is not clear what authors mean by the following and need changes Lines, 71, 72 “improvement of transposon reactivation” Line 72 “epigenetic alterations of RNA sequences”
Here we have done the requested changes.